

# Identification and functional analysis of *LecRLK* genes in *Taxodium* 'Zhongshanshan'

Jinbo Guo[1], Hao Duan[1], Lei Xuan[1], Ziyang Wang[1], Jianfeng Hua[1], Chaoguang Yu[1], Yunlong Yin[1], Mingzhi Li[2] and Ying Yang[1]

[1] Jiangsu Engineering Research Center for Taxodium Rich. Germplasm Innovation and Propagation, Institute of Botany, Jiangsu Province and Chinese Academy of Sciences (Nanjing Botanical Garden Mem. Sun Yat-Sen), Nanjing, China
[2] Genepioneer Biotechnologies Co. Ltd, Nanjing, China

## ABSTRACT

**Background**. Lectin receptor-like protein kinases (LecRLKs) can transform external stimuli into intracellular signals and play important regulatory roles in plant development and response to environmental stressors. However, research on the *LecRLK* gene family of conifers has seldom been reported.

**Methods**. Putative *LecRLK* genes were identified in the transcriptome of *Taxodium* 'Zhongshanshan'. The classification, domain structures, subcellular localization prediction, and expression patterns of *LecRLK* genes, as well as co-expressed genes, were analyzed using bioinformatics methods. Fifteen representative genes were further selected for qRT-PCR analysis in six tissues and under five different environmental stressor conditions.

**Results**. In total, 297 *LecRLK* genes were identified, including 155 G-type, 140 L-type, and 2 C-type. According to the classification, G-type and L-type *LecRLK* genes both can be organized into seven groups. The domain architecture of G-type proteins were more complex compared with that of L- and C-type proteins. Conservative motifs were found in G-type and L-type diverse lectin domains. Prediction and transient expression experiments to determine subcellular localization showed that LecRLKs were mainly concentrated in the cell membrane system, and some members were located at multiple sites at the same time. RNA-seq-based transcriptomics analysis suggested functional redundancy and divergence within each group. Unigenes co-expressed with *LecRLKs* in the transcriptome were found to be enriched in pathways related to signal transduction and environmental adaptation. Furthermore, qRT-PCR analysis of representative genes showed evidence of functional divergence between different groups.

**Conclusions**. This is the first study to conduct an identification and expression analysis of the *LecRLK* gene family in *Taxodium*. These results provide a basis for future studies on the evolution and function of this important gene family in *Taxodium*.

Corresponding author
Ying Yang, yingyang@cnbg.net, yo.ri@163.com

# INTRODUCTION

Throughout their life cycle, plants are exposed to constantly changing environments. The interaction between plant and environmental factors involves a series of signal perception and transduction pathways, including the receptor-like protein kinase (RLK) super family, which is a large cell–surface receptor family localized in the membrane (*Walker, 1994*). RLKs are composed of three parts: (i) an N-terminus extracellular domain, (ii) a transmembrane domain, and (iii) a C-terminal intracellular serine/threonine kinase domain. RLK proteins are divided into 15 families based on their extracellular domains (*Vaid, Macovei & Tuteja, 2013*). The lecRLK family is a group of RLKs containing extracellular lectin domains. Evidence acquired to date suggests that *LecRLK* family genes are present in many plant species and are thought to be plant-specific (*Navarro-Gochicoa et al., 2003*; *Vaid, Macovei & Tuteja, 2013*). The transmembrane domain and intracellular kinase domain of LecRLKs are highly conserved, whereas the variability of the extracellular lectin domain is high. The LecRLK family can be divided into three subgroups based on the extracellular lectin domain: G-type, L-type, and C-type (*Vaid, Macovei & Tuteja, 2013*). The G-type LecRLKs are proteins with an ectodomain that resembles the *Galanthus nivalis agglutinin* (GNA) mannose-binding motif (*Teixeira et al., 2018*). G-type LecRLKs were also called B-type LecRLKs, because GNA was first isolated from *Galanthus nivalis* bulbs, and B-lectin (PF01453) is the domain name of GNA in the Pfam database (*Teixeira et al., 2018*). The domain architecture of G-type LecRLK always contains a tandem repeat of a GNA (B-lectin) domain, an S-locus glycoprotein domain and/or a Pan/Apple (PNA) domain and/or a protein kinase domain (*Eggermont, Verstraeten & Van Damme, 2017*). The L-type (legume-like) LecRLKs are present in high amounts in legumes (*Eggermont, Verstraeten & Van Damme, 2017*). The domain architecture of L-type LecRLK includes a lectin-legB domain and/or a protein kinase domain (*Vaid, Macovei & Tuteja, 2013*). The extracellular lectin-legB domain contains a typical sandwich-fold structure in two complex topologies, suggesting that oligosaccharides may be the ligands (*Vaid, Macovei & Tuteja, 2013*). C-type (calcium-dependent) LecRLKs rely on $Ca^{2+}$ to function. They are abundant in mammals and participate in pathogen recognition and immune response. However, C-type LecRLKs seldom exist in plants, with studies only finding one C-type LecRLK in Arabidopsis (*Eggermont, Verstraeten & Van Damme, 2017*), rice, corn, *Populus* (*Yang et al., 2016*), and *Eucalyptus*, and two C-type LecRLKs in soybean (*Liu et al., 2018*) and wheat (*Triticum aestivum*) (*Shumayla et al., 2016*).

Plant *LecRLKs* can sense different external stimuli through the diverse extracellular domain and transform extracellular signals into intracellular signals through the intracellular kinase domain to facilitate the regulation of cell physiological and biochemical processes. The L-type *lectin receptor kinase-V.5* (*LecRK-V.5*) gene of Arabidopsis negatively regulates stomatal immunity; overexpressing LecRK-V.5 led to early stomatal reopening, making Arabidopsis more sensitive to pathogen invasion (*Desclos-Theveniau et al., 2012*). High levels of L-type lectin-like protein kinase 1 (AtLPK1) expression could promote seed germination and cotyledon greening under high salt stress. Additionally, expression of *AtLPK1* is reported to be strongly induced by abscisic acid (ABA), methyl jasmonate,

salicylic acid, and stress treatments (*Huang et al., 2013*). *Sun et al. (2013)* reported that ABA, salinity, and drought stress could all induce the expression of the *Glycine soja S-locus LecRLK* gene (*GsSRK*) in soybean, and *A. thaliana* transformed with GsSRK showed strong salt tolerance and high yield. In another study, the overexpression of *Pisum sativum LecRLK* in tobacco was shown to reduce ionic and osmotic components under salt stress so as to achieve salt-alkali tolerance (*Vaid et al., 2015*). L-type *LecRLKs* characterized from Antarctic moss (*Pohlia nutans*; *PnLecRLK1*) are reported to enhance the tolerance of *A. thaliana* to low temperatures and increase its sensitivity to ABA (*Liu et al., 2017*). In addition to the above functions, *LecRLKs* are important in plant growth and development. *LecRK IV.2* gene expression destroys pollen formation and leads to male sterility (*Wan et al., 2008*). The *A. thaliana LecRK-a1* gene is highly expressed in cells around injured areas, and *A. thaliana* L-type *LecRK-V.5* is presumed to play an important role in cell division (*Riou et al., 2002*). *GsSRK* was found to be involved in plant architecture (*Sun et al., 2018*), and the *Gossypium hirsutum GhlecRK* gene may play a role in fiber development (*Zuo et al., 2004*). The *LecRLK* family may play important roles in plant development, hormone regulation, and responses to biotic and abiotic stressors.

'Zhongshanshan' is the general term for the interspecific hybrid clones of bald cypress (*Taxodium distichum*) and Montezuma cypress (*T. mucronatum*). Several cultivars have been registered in China, such as 'Zhongshanshan 302' (*Wang et al., 2016*), 'Zhongshanshan 405' (*Yu, Xu & Yin, 2016*), and 'Zhongshanshan 406' (*Fan et al., 2018*). *Taxodium* species are all well-known water-tolerant pioneer trees and also have moderate salt tolerance and are native swamp forest tree species in coastal, brackish locations in the southern United States (*Allen, Chambers & Mckinney, 2015*). *Taxodium* species typically have long life spans, with many individuals found to have lived for more than 1,000–2,000 years, indicating that they have a strong resistance to pests and diseases (*Creech et al., 2011*). Therefore, *Taxodium* species have a strong tolerance to biotic and abiotic stressors (*Yang et al., 2018*). *Taxodium* 'Zhongshanshan' has been found to show strong adaptability in China, and is now widely used for wetland ecological restoration, coastal shelter forest construction, saline-alkali land afforestation, urban and rural afforestation, and farmland forest networks (*Changxiao et al., 2006*), offering great landscaping and ecological values. Recently, exploring the stress resistance of *Taxodium* at the molecular level has become a research focus (*Yu, Xu & Yin, 2016*; *Fan et al., 2018*; *Wang et al., 2017*; *Qi et al., 2014*). Because of their inability to move, plant can only passively withstand changes in the external environment and various stimuli. Therefore, receptors that can sense and specifically bind these stimuli, thus signaling molecules are very important for plants to adapt to environmental stressors. Study of the structure and function of the *LecRLK* gene family will be helpful to reveal the stress-related signal transduction network in *Taxodium* 'Zhongshanshan'.

At present, research on the *LecRLK* gene family in woody plants has only concentrated on a few tree species, like *Poplar* and *Eucalyptus* (*Yang et al., 2016*). Conifers are rarely reported due to the lack of genomic information. With the development of high-throughput sequencing technology, the transcriptome of *Taxodium* has been published (*Qi et al., 2014*; *Yu, Xu & Yin, 2016*; *Wang et al., 2019*), which has laid a foundation for the identification of large gene families. Here, we conducted a comprehensive bioinformatics analysis of

*ThzLecRLK* gene family based on transcriptome data and analyzed the expression of 15 representative *LecRLKs* in different tissues and under various stressors. This study provides insight for functional predictions of many members of the *LecRLK* gene family and provides a framework for further functional investigation of these genes.

## MATERIALS AND METHODS

### Identification of ThzLecRLK sequences

The unigene sequences of *Taxodium* 'Zhongshanshan' were derived from the previously determined salinity stress transcriptome (*Yu, Xu & Yin, 2016*), short-term waterlogging transcriptome (*Qi et al., 2014*), and adventitious roots transcriptome (*Wang et al., 2019*). Hidden Markov Model (HMM) profiles (PF00069, PF01453, PF00139, PF00059), which correspond to kinase, G-type lectin, L-type lectin, and C-type lectin domains, respectively, were downloaded from the Pfam database (http://pfam.xfam.org/) (*El-Gebali et al., 2019*). We retrieved genes containing a kinase domain by running the hmmsearch program (HMMER 2.3.2) to search the kinase profile (PF00069) against the proteomic sequences. Within this set of hypothetical kinase proteins, we searched for G-type lectin, L-type lectin, and C-type lectin HMM profiles (PF01453, PF00139, PF00059) (*E* value cut-off < 1). Sequences in which we identified a protein kinase domain (PF00069), along with either a G-type lectin (PF01453), an L-type lectin (PF00139), or a C-type lectin domain (PF00059) were selected. Pfam and NCBI CCD (Conserved Domain Database) (https://www.ncbi.nlm.nih.gov/cdd/) were further used to check their kinase domain and corresponding lectin domains to determine the putative lecRLKs. ThzLecRLK genes were finally confirmed after removing redundant sequences with 97% similarity between different database.

### Sequence alignment, classification, and subcellular localization classification

Multiple alignments of the nucleotide and amino acid sequences were performed using MAFFT v7 (*Katoh & Standley, 2013*), and the auto alignment mode was selected. The classification of group and subgroups was constructed based on the sequences of ThzLecRLK proteins from *A. thaliana* and *Taxodium* 'Zhongshanshan' using a neighbor-joining (NJ) method with 1,000 bootstrap replicates. ClustalX2.1 was used to construct NJ tree, and MEGA7.0 software (http://www.megasoftware.net/) (*Kumar, Stecher & Tamura, 2016*) was used for beautification. C-type proteins were used as the outgroup for the subgroup classification of G-type and L-type lecRLKs.

The protein theoretical molecular weight and theoretical isoelectric point (pI) were predicted using compute pI/MW (http://au.expasy.org/tools). The subcellular localization of PtAspAT proteins was predicted using the PSORT (https://psort.hgc.jp/) and CELLO programs (http://cello.life.nctu.edu.tw/). In addition, ThzlecRLK-C-type-2 and ThzlecRLK-L-type-86 were selected for a transient expression experiment. The coding DNA sequence (CDS) regions were inserted into a 16318-hGFP expression vector (Xian Chunfeng Biotechnology Co., Ltd). The transient expression vectors (35S::ThzlecRLK-C-type-2-GFP and 35S::ThzlecRLK-L-type-86-GFP) containing green-fluorescence protein

(GFP) were transferred into the protoplast of *A. thaliana* following the method of *Yoo, Cho & Sheen (2007)*. The fluorescent signals were observed with a TCS SP8 DIVE Confocal laser scanning microscope (Leica, Germany).

## Analysis of the protein domain of ThzLecRLK sequences

The Pfam 32.0 program (*El-Gebali et al., 2019*), NCBI CDD and TMHMM Server v. 2.0 (http://www.cbs.dtu.dk/services/TMHMM/) were used to statistically identify conserved domain and transmembrane helices (TMhelix) in the complete amino acid sequences of ThzLecRLK proteins. The positions of all conserved domains and TM were marked and shown in a diagram based on their physical location. The sequences of the conserved domain were generated using the online Weblogo platform (http://weblogo.berkeley.edu/) (*Crooks et al., 2004*).

## Transcriptional profile analysis

For *ThzLecRLK* gene expression analysis, RNA-seq data of different tissues and treatments of *Taxodium* 'Zhongshanshan' were obtained from the pre-laboratory development of the stress transcriptome (*Yu, Xu & Yin, 2016*) and short-term waterlogging transcriptome (*Qi et al., 2014*). The waterlogging stress related transcriptome analysis included four groups of data: the untreated root, the untreated stem, the root after one hour of soil waterlogging, and the stem after one hour of soil waterlogging. The salt stress related transcriptome analysis included root data under four treatments, namely, untreated root (T1), 100 mM NaCl treatment for 1 h (T2), 200 mM NaCl treatment for 1 h (T3), and 200 mM NaCl treatment for 24 h (T4). The level of gene expression was calculated by determining the number of expected fragments per kilobase of transcript per million mapped reads (FPKM). Heatmaps were generated using the heatmap package in R version 3.4.0 with log2 (FPKM+1) value.

## Co-expression gene analysis

The eight stress conditions mentioned above, i.e., four waterlogging and four salt stress transcriptomes, were analyzed to determine unigenes co-expressed with LecRLKs. All the unigenes with FPKM $\geq$1 were included to calculate the correlation coefficient. The R package 'psych' was employed to determine the Pearson's correlation coefficient (PCC) values between ThzLecRLK unigenes and other unigenes. Correlation pairs were deemed statistically significant when the |PCC| was >0.9 and the corrected p-vale (FDR method) was <0.05. The co-expressed genes were used to analyzed the hypergeometric Fisher exact test ($P < 0.05$) and Benjamini correction (FDR < 0.05) to construct the map of enriched KEGG pathways. The transcription factors (TFs) in the co-expression genes were identified by searching the plant transcription factor database PlantTFDB 4.0 with all assembled unigenes (*Jin et al., 2017*).

## RNA extraction qRT-PCR analysis

In September 2018, the clones of one-year-old *Taxodium* 'Zhongshanshan 406' were taken, the roots were washed and put into water for hydroponic growth for a month. Then plants were treated with ABA (1 mM), SA (1 mM), mannitol (200 mM), NaCl

(300 mM), and 4 °C treatment respectively. Shoots were collected from the seedlings subjected to all treatments at 6 h. Shoos untreated with any stresses were taken as control. For organ-specific expression, Roots, stems, phloem, xylem and leaves were collected from clones of one-year-old *Taxodium* 'Zhongshanshan 406' and fruits were taken from mature clones of 'Zhongshanshan 406'. Root samples were taken as control. Three different plants as three biological replicates were randomly harvested. All collected samples were frozen in liquid nitrogen immediately after harvest and stored at −80 °C until RNA extraction.

Total RNA was extracted using an RNeasy® Plant Mini Kit (QIAGEN, Shanghai, China). First-strand cDNA was synthesized using an HiScript II Q RT SuperMix for qRT-PCR (+g DNA wiper) (Vazyme Biotechnology Co., Ltd, Nanjing, China). Primers for qRT-PCR were designed using Primer software (https://www.genscript.com/tools/pcr-primers-designer) (File S1). SYBR Green Reagents were used to detect the target sequence. PCR amplifications were performed in a StepOnePlus real-time thermal cycler (Thermo Fisher Scientific, San Jose, CA, USA) in a final volume of 20 µL containing 2.0 µL of cDNA, 0.4 µL of each primer (200 nM), 7.2 µL of sterile water, and 10 µL (2×) of SYBR Green qRT-PCR Master Mix Kit (Bimake, Nanjing, China). Adenine phosphoribosyl transferase was used as the control to minimize variations in the cDNA template. Data represent three biological replicates and three technical replicates. The conditions for amplification were 10 min of denaturation at 95 °C , followed by 40 cycles of 95 °C for 15 s, 60 °C for 30 s, and 72 °C for 30 s, after which a melt curve was produced at 60 ∼95 °C . (*Ma et al., 2018*). The relative expression levels of genes were calculated using the $2^{-\Delta\Delta Ct}$ method (*Xia et al., 2014*).

## RESULTS

### Identification and classification of LecRLKs in Taxodium 'Zhongshanshan'

Using the unigene functional annotation database of the *Taxodium* 'Zhongshanshan' transcriptome, 1,331 unigenes were predicted to contain a protein kinase domain (PF00069). After removing incomplete open reading frame (ORF) sequences, 446 unigenes remained. Online SMART (http://SMART.embl-heidelberg.de) software was used to view the schematic map of protein structures, remove sequences without transmembrane domain (TM) and LecRLK domains. Ultimately, a total of 297 unigenes were obtained (File S2). All LecRLK proteins were classified into G-type, L-type, and C-type according to the structure of the extracellular lectin domain. In total, 155 G-type, 140 L-type, and two C-type LecRLKs were identified (Fig. 1, File S3).

### Classification of ThzLecRLKs

A NJ tree was generated using multiple alignments of the complete LecRLK protein sequences of *Taxodium* 'Zhongshanshan' and *A. thaliana*. The cladograms of G-type and L-type LecRLKs were constructed separately and clustered into subgroups according to homologous relationships (Fig. 2). The cladogram of the 155 G-type members of the LecRLK family proteins of *Taxodium* 'Zhongshanshan' were divided into seven clusters. Some of these groups were then subdivided, based on distinct clade formation. Group II, IV and VII was subdivided into three subgroups (II-a to II-c, IV-a to IV-c, VII-a to VII-c) , and

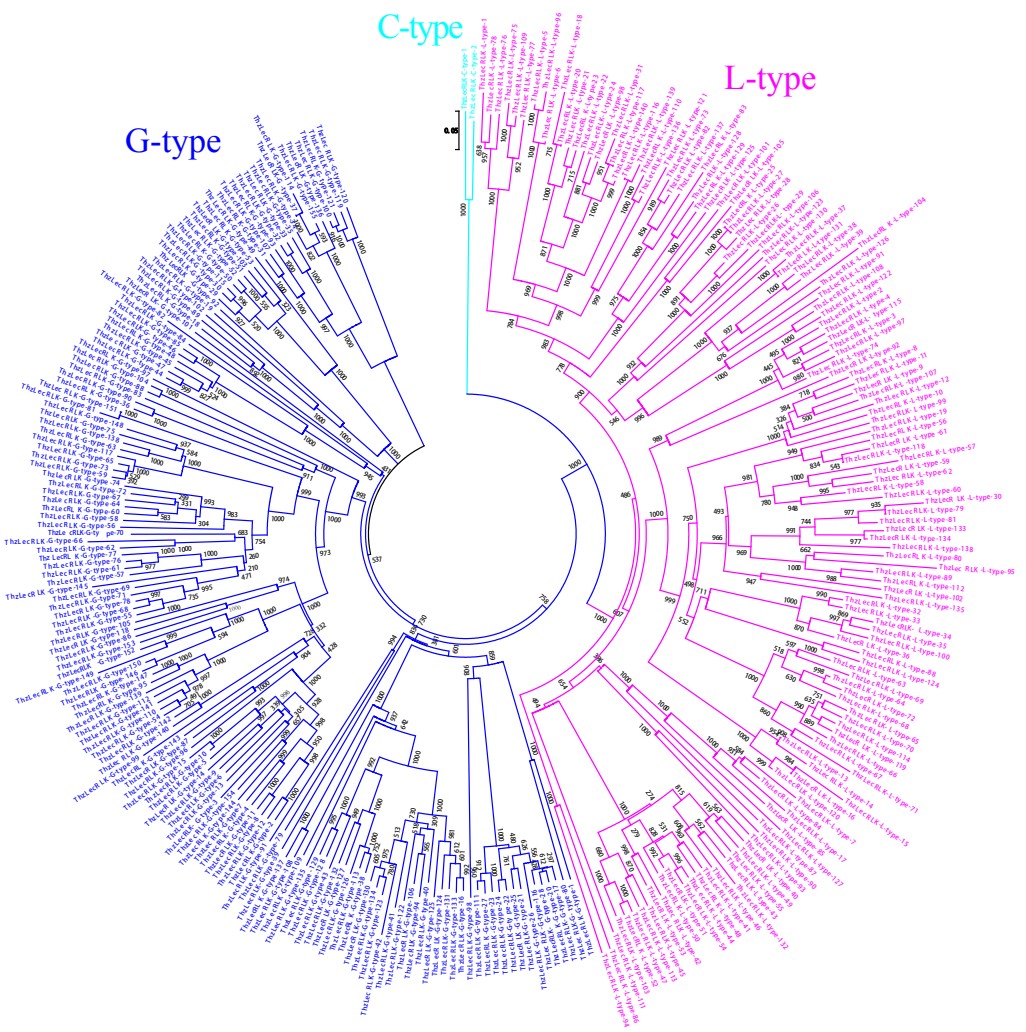

**Figure 1** **Classification of 297 LecRLKs genes.** The cladogram was constructed by neighbor-joining (NJ) method. Note that three different types of ThzLecRLKs are categorized clearly in three different clades (blue, G-type; cyan: C-type; fuchsia: L-type).

Group VI was subdivided into five subgroups (V-a to V-e). Most *AtLecRLKs* were assigned to the same subgroup with only At2G19130.1 being assigned to the Group VI-a subgroup and At4G32300.1 to the Group IV-a clade (Fig. 2A, File S4). Similarly, for the L-type LecRLKs, the cladogram was also divided into seven clusters. Group II was subdivided into seven subgroups (II-a to II-g), Group V was subdivided into four subgroups (V-a to V-d), and Group VI was subdivided into five subgroups (VI-a to VI-e). Each group, except the singleton Group V and VII, contained at least one *A. thaliana* L-type LecRLK (Fig. 2B, File S5). The division of each group was supported by high bootstrap values.

## Domain architecture analysis

Predictive domain architecture and organization analysis assisted in the discovery and characterization of the conserved domain of ThzLecRLKs (Fig. 3). The domain architecture

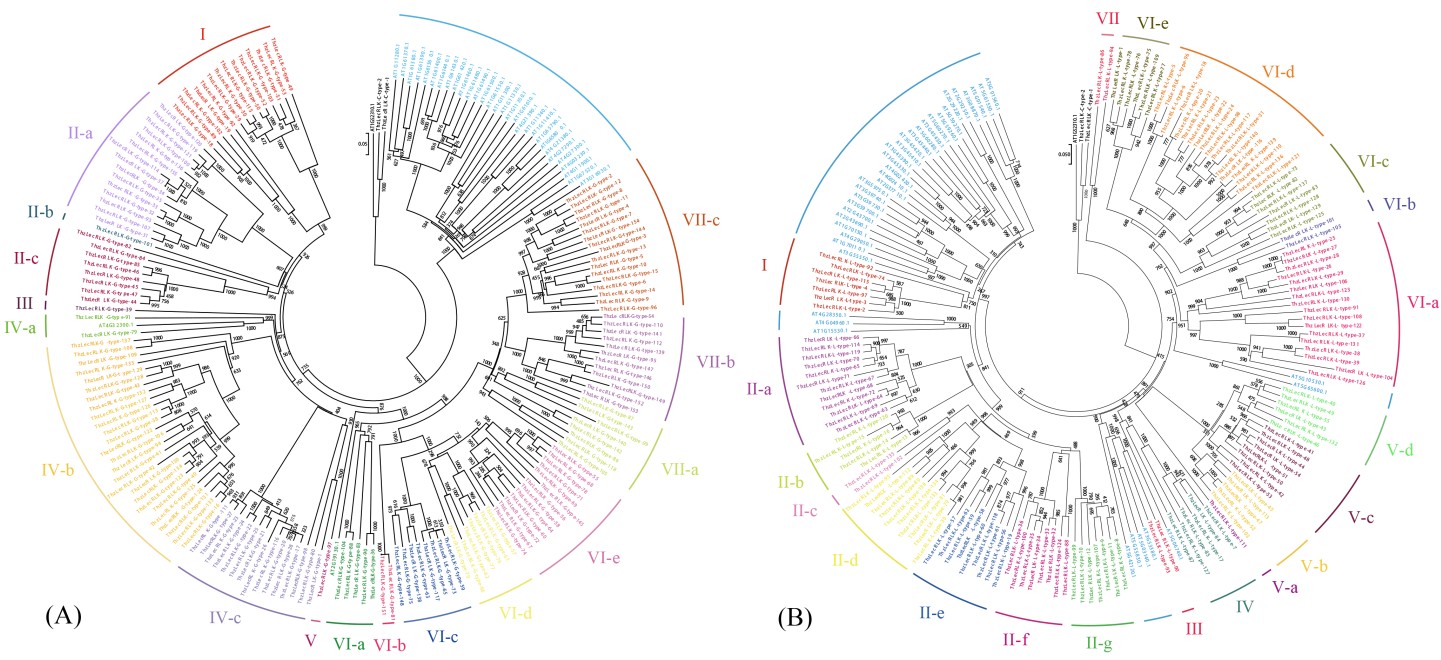

**Figure 2** **Classification of different groups of G and L-type ThzLecRLKs.** Different color regions are used to distinguish different subgroups. (A) Cladogram of G type LecRLK gene family. The neighbor-joining (NJ) method was used to analyze the evolutionary trees of 155 G Thzlecrks and 32 G AtLecRLK. (B) Cladogram of L type LecRLK gene family. The cladogram of L-type ThzLecRLK analyzed 50 L-type ThzLecRLK and 39 AtLecRLK with NJ.

of G-type LecRKs were known to always have a tandem repeat of a GNA (B_lectin) domain, an S-locus glycoprotein domain and/or a PNA domain and/or a protein kinase domain (Fig. 4A). Within the 155 G-type ThzLecRLKs, 87 contained all four basic domains (kinase, B-lectin, S-locus glycoprotein, PAN), 25 contained kinase, B-lectin, and S-locus glycoprotein, 15 contained kinase, B-lectin and PAN, and the remaining 28 ThzLecRLKs only had kinase and B-lectin domains (File S6). In addition, an APH domain was found in the ThzLecRLK-G-type-109 kinase domain, and a PIP49_C domain was identified in the ThzLecRLK-G-18 kinase domain. Compared with G-type proteins, the domain architectures of L- and C-type ThzLecRLKs were less complex. The L-type ThzLecRLK agglutinin domain contained only a lectin_legB domain and kinase domain (Fig. 4B). Both C-type ThzlecRLKs are composed of a lectin-C domain and kinase domain (Fig. 4C). All of the ThzLecRLKs were predicted to contain at least one TM domain, with 187 (G-type:96, L-type:90, C-type:1) ThzLecRLKs containing only one TM domain, 93 (G-type:43, L-type:49, C-type:1) containing two TM domains, and 17 (G-type:16, L-type:1) containing three TM domains.

To investigate the presence of conserved motifs in the main domains of ThzLecRLK, the amino acid sequences of intracellular kinase domains of all ThzLecRLKs, G-type ectodomains (B_lectin domain, S-locus glycoprotein domain and PNA domain), and L-type ectodomains (lectin_legB domain) were aligned. For the kinase domains, the alignment revealed the overall conservation of the ATP binding (consensus motif GxGxxGxV) and the

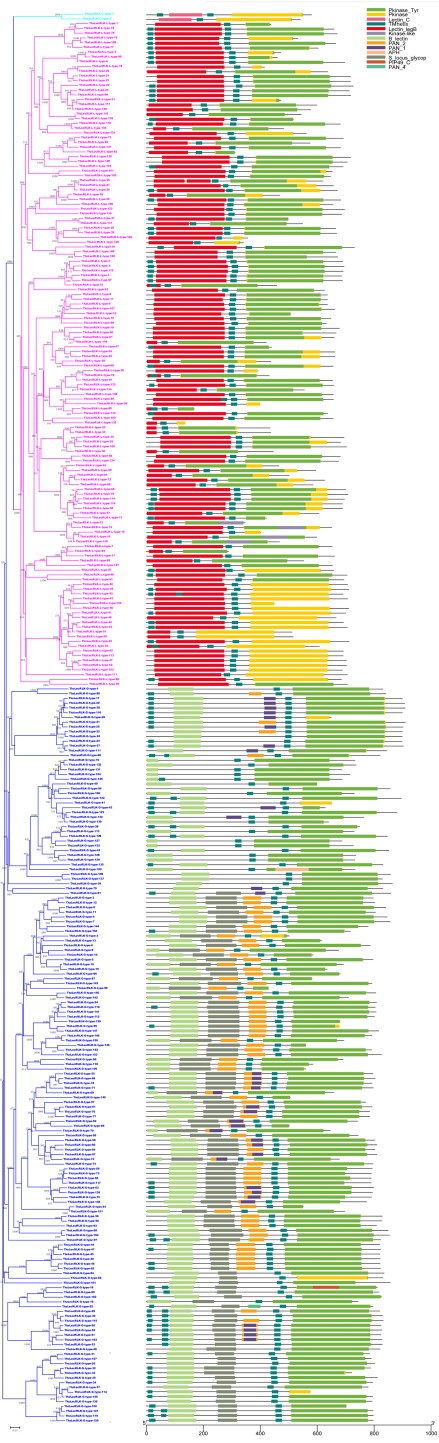

**Figure 3** **Conserved domain in lectin domain of ThzLecRLK (total 297) proteins.** C-type, conserved domain in lectin domain of C-type ThzLecRLKs (total two) protein; L-type, conserved domain in lectin domain of L-type ThzLecRLKs (total 140) protein; G-type, conserved domain in lectin domain of G-type LecRLKs (total 155) protein.

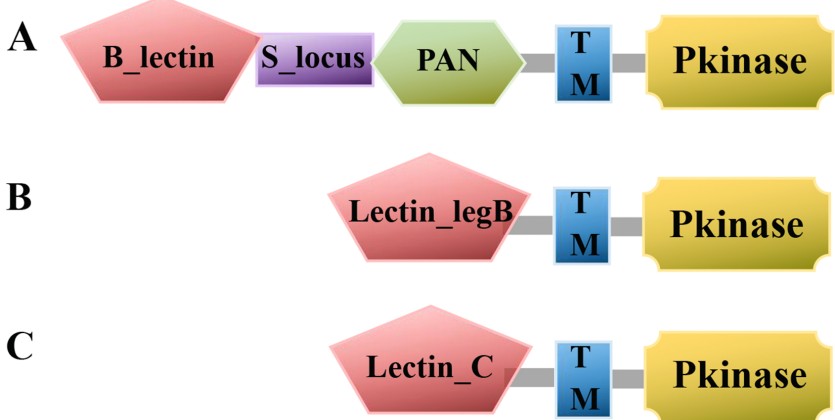

**Figure 4** **Line model of the three LecRLK classes.** The LecRLK family can be divided into three subgroups based on the extracellular lectin domain: G-type, L-type, and C-type. (A) The domain architecture of L-type LecRLK includes a lectin-legB domain and/or a protein kinase domain(Pkinase). (B) The domain architecture of G-type LecRLK always contains a tandem repeat of a GNA (B-lectin) domain, an S-locus glycoprotein domain and/or a Pan/Apple (PNA) domain and/or a protein kinase domain. (C) C-type LecRLKs characterized by the presence of calcium-dependent carbohydrate-binding domains.

catalytic sites (consensus motif HxDxKxxN) (marked with red boxes) (File S7) (*Hanks & Hunter, 1995*). Two conversed motifs were identified in the S-locus glycoprotein domain, including a GYM element (65 G, 66 Y, 67 M) and a cysteine-rich (C-rich) motif. The C-rich motif was similar to that of the poplar EGF domain (*Yang et al., 2016*). Since the epidermal growth factor (EGF) domain was not predicted by the software used here, the C-rich motif could overlap the sequence of the N-terminal region of the S-locus glycoprotein and the C-terminal region of EGF domains. Three highly conserved motifs were found in the PNA domain of G-type ThzLecRLKs, including an FFS element (18 F, 19 F, 20 S), RYDVS element (59 R, 60 Y, 61 D, 62 V, 63 S, 64 D), and another C-rich motif. The C-rich motif in the middle of the PNA domain was also found to be highly conserved in poplar (*Yang et al., 2016*), Arabidopsis, and tomato G-type LecRLKs (*Teixeira et al., 2018*). Three elements (WQWSD, NxxSKRK, and WxxLxxP) were found to be conversed in B-lectin domains. In addition, four conversed motifs were found within the lectin-leg B domain of L-type PtLecRLKs.

## Physical and chemical properties and prediction of subcellular localization

After determining the conserved domains of ThzLecRLKs, we predicted the physicochemical properties and subcellular localization of these proteins. The results showed that 297 ThzLecRLKs had a predicted ORF sequence length ranging from 136 to 910 amino acids, a molecular weight ranging from 15.24 to 101.24 kDa, and pI value ranging from 4.90 to 9.56 (File S2).

PSORT and CELLO were used for the subcellular localization prediction, and 266 ThzLecRLKs had common predicted localizations by both methods. Of these, 213 ThzLecRLKs were located in the plasma membrane and 28 were located at more than

**Table 1   Prediction of subcellular location of 297 ThzLecRLKs.**

| Type | Number | Percentage(%) | Type | Number | Percentage(%) |
|------|--------|---------------|------|--------|---------------|
| PM | 213 | 71.7 | Extr+nucl | 1 | 0.3 |
| chlo | 9 | 3 | PM+extr | 7 | 2.4 |
| cyto | 6 | 2 | PM+chlo | 8 | 2.7 |
| extr | 3 | 1 | PM+nucl | 2 | 0.7 |
| mito | 1 | 0.3 | PM+cyto | 2 | 0.7 |
| nucl | 6 | 2 | PM+mito | 1 | 0.3 |
| cyto+chlo | 5 | 1.7 | PM+Vacu | 1 | 0.3 |
| cyto+ER | 1 | 0.3 | No | 31 | 10.4 |

**Notes.**

Only the consistent prediction results of the two softwares were analysed.

No, the prediction results of the two softwares were not consistent; PM, plasma membrane; chlo, chloroplast; vacu, vacuole; extr, extrocytoplasmic surface; nucl, nucleus; ER, endocytoplasmic reticulum; cyto, cytoplasm; mito, mitochondrion.

one site. The remaining nine were located in the chloroplasts, three in the extracytoplasmic surface, six in the nucleus, six in the cytoplasm and one in the mitochondrion (Table 1; File S8). In addition, two genes were selected for an instantaneous expression experiment to further explore the of subcellular localization characteristics of the LecRLK family. Fluorescence signals of ThzlecRLK-C-type-2 were observed on the plasma membrane and nucleus (Fig. 5). Fluorescence signals of ThzlecRLK-L-type-86 were observed on the plasma membrane, chloroplast, and nucleus (Fig. 6).

## Transcriptome heat map analysis

As the first attempt to provide insight into the potential function of *ThzLecRLK* genes, we analyzed the expression of the 297 *ThzLecRLK* genes under salt and waterlogging stress by mining previously published transcriptome data of *Taxodium* 'Zhongshanshan' (*Yu, Xu & Yin, 2016*; *Qi et al., 2014*). Under salt stress, 241 *ThzLecRLKs* (G-type: 129; L-type: 111; C-type: 1) were not detected (FPKM = 0) in any sample. We generated a heat map of the 56 (G-type:26; L-type: 29; C-type: 1) expressed *ThzLecRLK* genes (FPKM>0) using the FPKM values (Fig. 7A). Many genes belonging to the same groups were clustered together on the heat map, such as G-type *ThzLecRLKs* from group I (*ThzLecRLK-G-type-29*, *ThzLecRLK-G-type-30*, and *ThzLecRLK-G-type-115*) and group VI-c (*ThzLecRLK-G-type-59* and *ThzLecRLK-G-type-117*), and group VII-b (*ThzLecRLK-G-type-54* and *ThzLecRLK-G-type-112*), L-type *ThzLecRLKs* from group VI-d (*ThzLecRLK-L-type-117*, *ThzLecRLK-L-type-18*, and *ThzLecRLK-G-type-31*), group III (*ThzLecRLK-L-type-90*, and *ThzLecRLK-L-type-93*), and group II-e (*ThzLecRLK-L-type-118* and *ThzLecRLK-L-type-19*).

Under waterlogging stress, 241 *ThzLecRLKs* (G-type: 130; L-type: 110; C-type: 1) were not detected in any sample. The expression of the remaining 56 *ThzLecRLKs* (FPKM ≥ 1) under waterlogging stress were analyzed (Fig. 7B). Genes from the same group, such as L-type *ThzLecRLKs* from group VI-d (*ThzLecRLK-L-type-110* and *ThzLecRLK-L-type-96*), and group VI-a (ThzLecRLK-L-type-130 and ThzLecRLK-L-type-123) and G-type *ThzLecRLKs* from group IV-c (*ThzLecRLK-G-type-80*, and *ThzLecRLK-L-type-98*) were also clustered close together on the heat map.

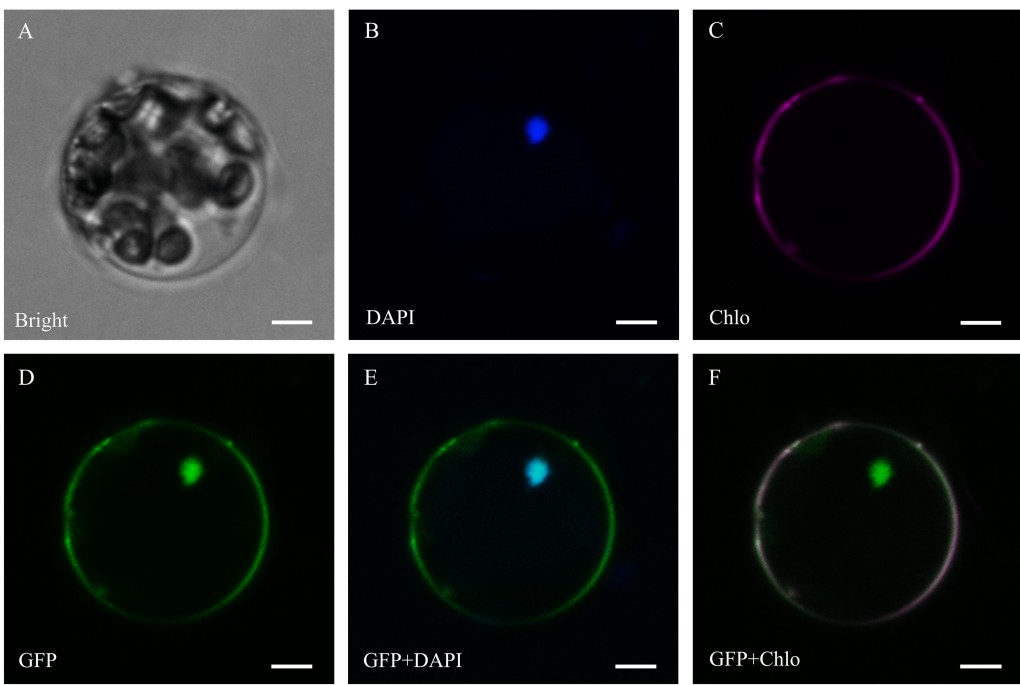

**Figure 5 Subcellular localization of ThzLecRLK-C-type-2 in *Arabidopsis thaliana*.** (A) Bright; (B) DAPI, the protoplasts are stained with DAPI to visualize the nucleus; (C) Chlo, Chloroplast autofluorescence (type2 Dil staining); (D) GFP, fluorescence of ThzLecRLK-L-type-86-GFP; (E) GFP+DAPI, merged images of GFP and DAPI ones; (F) GFP+Chlo, merged images of GFP and Chlo ones.

In total, 87 *ThzLecRLKs* (G-type:42; L-type: 43; C-type: 2) were expressed under the two different stress conditions. Among them, 31 *ThzLecRLKs* (G-type:17; L-type: 13; C-type: 1) were only expressed under salt stress and another 31 *ThzLecRLKs* (G-type:16; L-type: 14; C-type: 1) were only expressed under waterlogging stress. The remaining, 25 *ThzLecRLKs* (G-type: 9; L-type: 16) were found to potentially be involved in both stress responses.

## Co-expression analysis

More than 9,000 co-expressed genes of the 87 *ThzLecRLKs* with RKPM > 0 were identified. Among them, 230 transcription factors (TFs) were predicted to be co-expressed with 20 *ThzLecRLKs* (G-type 6, L-type 14), indicating that there may be regulatory relationships between these *ThzLecRLKs* and TFs. Interesting, the 20 *ThzLecRLKs* were among those expressed under both stress conditions. The co-expressed TFs were mainly MYB (28), ERF (19), $C_2H_2$(19), bHLH (14), GRAS (10), and NAC (9) (Fig. 8). ThzLecRLK-L-type128 had the greatest number of TFs (82), followed by ThzLecRLK-L-type86 (52) and ThzLecRLK-L-type88 (46) (File S9).

The enrichment pathway ($P < 0.05$) of co-expressed genes was analyzed using the Kyoto Encyclopedia of Genes and Genomes (KEGG). The most enriched pathways included folding, sorting, and degradation, environmental information processing, signal transduction, environmental adaptation, and plant pathogen interaction (Fig. 9). These

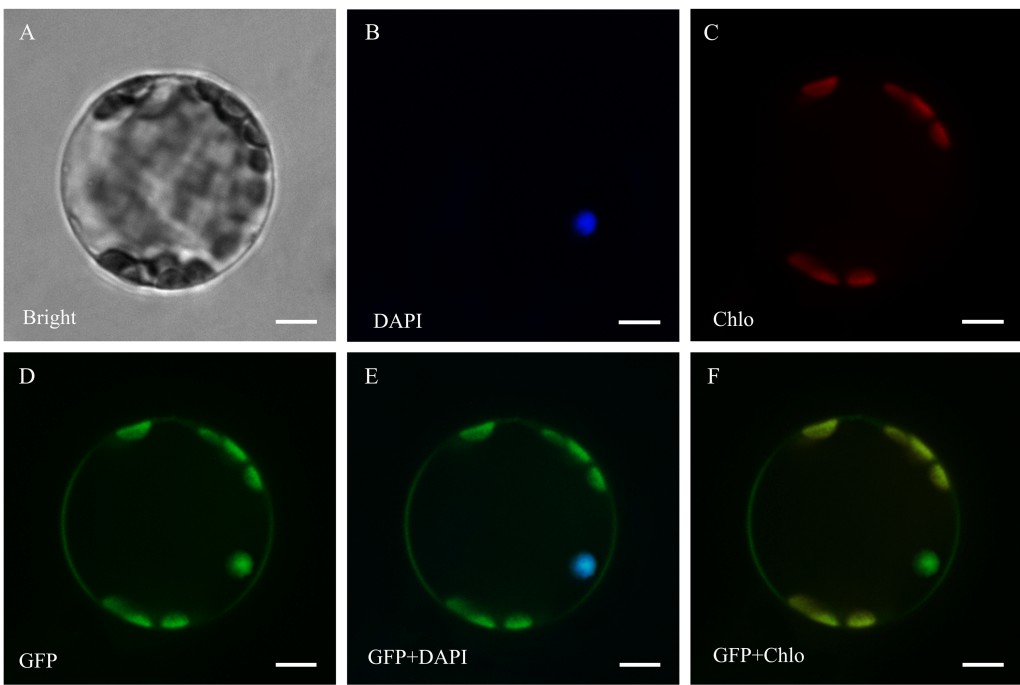

**Figure 6** Subcellular localization of ThzLecRLK-L-type-86 in *Arabidopsis thaliana*. (A) Bright; (B) DAPI, the protoplasts are stained with DAPI to visualize the nucleus; (C) Chlo, Chloroplast autofluorescence; (D) GFP, fluorescence of ThzLecRLK-L-type-86-GFP; (E) GFP+DAPI, merged images of GFP and DAPI ones; (F) GFP+Chlo, merged images of GFP and Chlo ones.

results indicated that the *ThzLecRLK* genes were closely related to signal transduction, stress response, and other biological pathways.

## Differential expression of LecRLKs in various tissues

In order to determine whether diversity in structure led to varying expression patterns in tissues (i.e., roots, stems, xylem, phloem, leaves, and fruits), we selected 15 genes (G-type: 7; L-type: 6; C-type: 2) to be subject to qRT-PCR. We selected one gene from each group of G-type and L-type as representative genes, and all of them were found to be stress-responsive genes revealed by the RNA-seq-based transcriptomics analysis (Table 2). Of the 15 selected *ThzLecRLK* genes, six (G-type:3 and L-type:3) showed peak expression levels in roots and four (G-type:1 and L-type:3) showed peak expression levels in phloem (Fig. 10). In addition, the G-type genes *ThzLecRLK-G-type-97* and *ThzLecRLK-G-type-98* exhibited peak expression levels in the stem and xylem, respectively; however, *ThzLecRLK-G-type-30* showed peak expression levels in leaves. The L-type gene *ThzLecRLK-L-type-109* exhibited high levels of expression in both stem and phloem tissues. The expression of *ThzLecRLK-C-type-1* was the highest in fruit and the lowest in the stem, while the expression of *ThzLecRLK-C-type-2* was the highest in leaves and lowest in xylem.

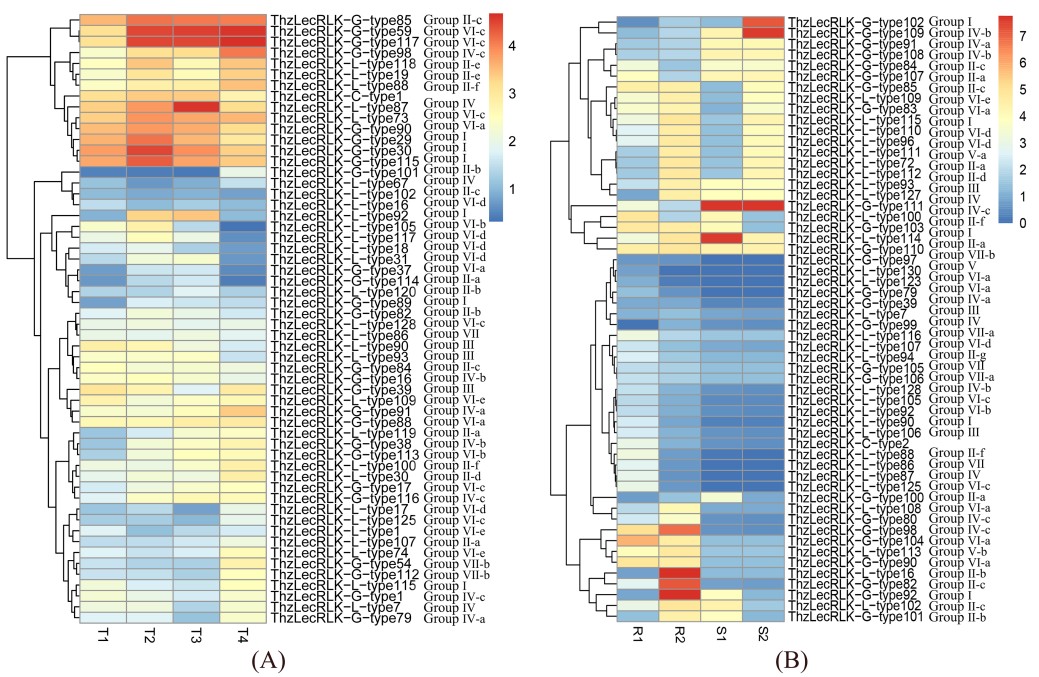

**Figure 7 Expression patterns of ThzLecRLK genes.** RNA-seq data of different tissues and treatments of T. hybrid 'Zhongshanshan' were obtained from NCBI SRA (Sequence Read Archive). The specificity of gene expression was determined by comparing FPKM values in specific treatments. The classifications of genes in phylogenetic trees were listed next to the gene name. (A) The FPKM value of four different Nacl treatment including T1 (Untreated), T2 (Roots treated with 100 mM Nacl for 1 hour), T3 (Roots treated with 200 mM Nacl for 1 hour) and T4 (Roots treated with 200 mM Nacl for 24 h) were analyzed. (B) The FPKM value of four different waterlogging treatment including R1 (Untreated root), R2 (Root after 1 hour of waterlogging), S1 (Untreated leaves) and S2 (Root after 1 hour of leaves) were analyzed.

## Differential expression of LecRLKs under a variety of stressors

To investigate the potential roles that *LecRLK* genes may play in different stress response mechanisms of *Taxodium* 'Zhongshanshan', seedlings were treated with ABA, salicylic acid (SA), mannitol, NaCl, and low temperature (4 °C) (Fig. 11). Of the 15 genes screened, ten (G-type:4, L-type:4 and C-type:2) showed peak expressions at 4 °C. In addition, *ThzLecRLK-G-type-90* showed the highest relative expression under exposure to ABA, and *ThzLecRLK-G-type-91* showed the highest relative expression under mannitol treatment. We also found that the relative expression of *ThzLecRLK-G-type-101* gene under ABA, SA, mannitol, NaCl, and 4 °C treatments decreased significantly compared with that in the nutrient water treatment. As for the L-type genes, *ThzLecRLK-L-type-87* showed the highest relative expression under NaCl treatment and *ThzLecRLK-L-type-109* showed the highest relative expression under exposure to ABA. Of these differentially expressed genes, *ThzLecRLK-G-type-30, ThzLecRLK-G-type-98, ThzLecRLK-L-type-93*, and *ThzLecRLK-L-type-111* were significantly induced by low temperature stress and their relative expression levels were 5.7, 7.4, 6.3, and 19 times higher than in the controls, respectively.
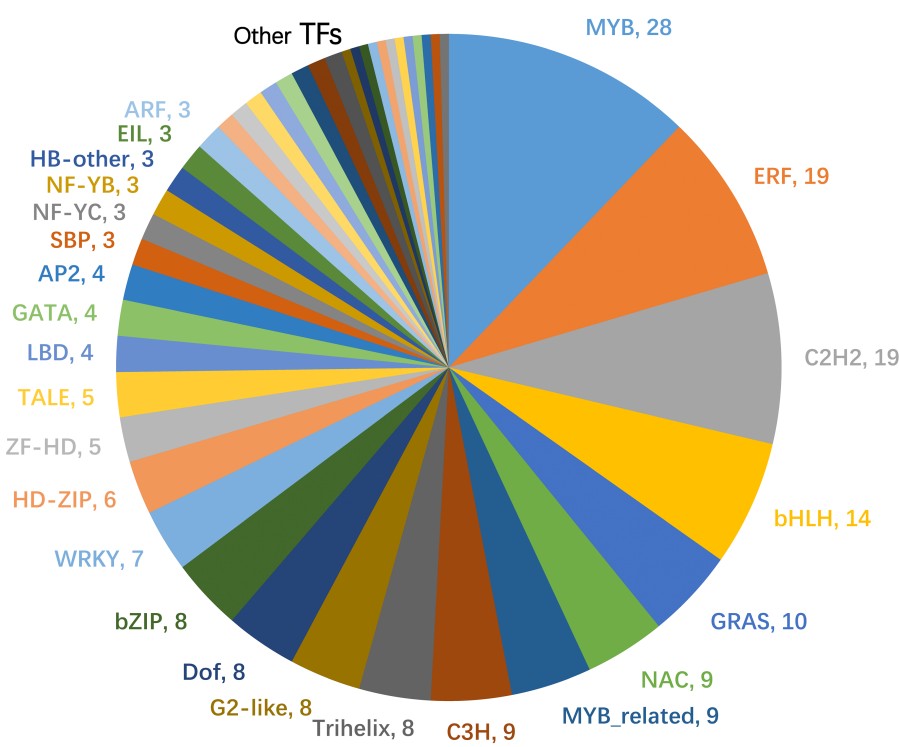

**Figure 8** **Statistics of the number of transcription factors predicted to be co-expressed with** *ThzLecR-LKs.* The number of each type was listed next to the abbreviation of the transcription factor.

## DISCUSSION

### Identification and classification of ThzLecRLKs

LecRLK family proteins were found to be distributed in all plants, with the number ranging from 21 to 325 in Ma's research (*Ma et al., 2018*). However, there was no correlation between the number of lecRLK genes and the genome size of species (*Ma et al., 2018*). In this paper, 297 *LecRLK* genes were identified from the transcriptome of *Taxodium* 'Zhongshanshan'. As these proteins are heavily involved in plant growth, development, and stress tolerance, the large number of LecRLKs in *Taxodium* 'Zhongshanshan' may indicate that *Taxodium* have evolved a large number of *LecRLKs* to respond to a longer life cycle and greater likelihood of exposure to more diverse external stimuli (*Liu et al., 2018*). We found 140 L-type, 155 G-type, and two C-type *ThzLecRLKs*. All embryonic plants were found containing three subgroups of lecRLKs (L, G and C). The number of G-type lecRLK is more than that of L-type lecRLK in most species, except Arabidopsis, *Capsella rubella* (*Ma et al., 2018*), shrub and corn (*Yang et al., 2016*). *Wan et al. (2008)* considered that cross-pollinated plants contains more G-type proteins than self-pollinated plants like *A. thaliana*; this may be due to the fact that G-type LecRLKs contain a special S-locus glycoprotein domain that regulates the self-incompatibility of pollen. There are generally 1-3 C-type *LecRLKs* in plants, indicating the relative stability of C-type *LecRLKs* in both gymnosperms and angiosperms. Finding at least one C-type in each species suggested the

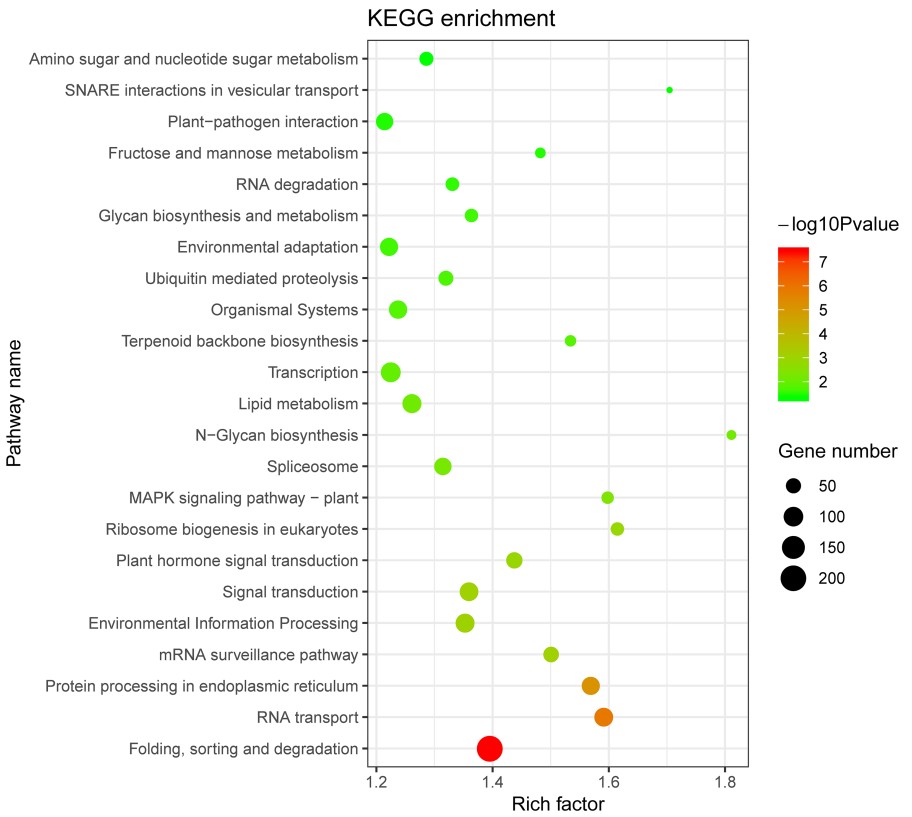

**Figure 9** **The 23 top Kyoto Encyclopedia of Genes and Genomes (KEGG) enrichment pathways of the unigenes co-expressed with LecRLK genes.**

indispensable role of C-type *LecRLKs* in plants. The number of G-type *ThzLecRLKs* was similar to most reported plants (*Ma et al., 2018*), ranged from 100 to 200. The number of L-type *ThzLecRLKs* was much higher than other species, which were all reported to be less than 100. Taken together, these results suggested that the expansion of the *LecRLK* gene family in *Taxodium* compared to angiosperms may related to the disproportionate expansion of L-type LecRLKs.

## Classification of ThzLecRLKs

The classification of G-type ThzLecRLKs and AtLecRLKs indicated that 30 out of the 32 G-type AtLecRLKs were assigned to independent clusters. Similarly, for L-type, 33 out of the 42 L-type AtLecRLKs were clustered into independent branches. According to the NJ tree of G-type and L-type ThzLecRLKs and AtLecRLKs, most sequences from the two species are separate, with only a few sequences from Arabidopsis clustered together with the sequences of *Taxodium* 'Zhongshanshan'. This suggested that ThzLecRLKs may have different evolutionary functions from similar proteins in *A. thaliana*. The same trend has also been observed between other woody and herbaceous plants (*Yang et al., 2016*). Specifically, the LecRLKs between different woody species formed clades separate from

**Table 2  Physical and chemical properties and subcellular localization prediction of 15 *ThzLecRLKs*.**

| Unigene number | Classfication | ORF length (aa) | Theoretical pI | Molecular weight (kDa) | Location |
|---|---|---|---|---|---|
| ThzLecRLK-G-type-101 | Group II-b | 856 | 6.71 | 95.21 | Plasma membrane |
| ThzLecRLK-G-type-85 | Group II-c | 822 | 6.24 | 90.53 | Plasma membrane |
| ThzLecRLK-G-type-30 | Group I | 829 | 5.44 | 93.08 | Plasma membrane |
| ThzLecRLK-G-type-97 | Group V | 823 | 8.96 | 90.33 | Plasma membrane |
| ThzLecRLK-G-type-90 | Group VI-a | 827 | 6.74 | 90.45 | Plasma membrane |
| ThzLecRLK-G-type-91 | Group IV-a | 858 | 6.05 | 94.58 | Plasma membrane |
| ThzLecRLK-G-type-98 | Group IV-c | 821 | 5.54 | 90.30 | Plasma membrane |
| ThzLecRLK-L-type-115 | Group I | 691 | 6.31 | 76.95 | No |
| ThzLecRLK-L-type-88 | Group II-f | 681 | 6.44 | 76.32 | Plasma membrane |
| ThzLecRLK-L-type-93 | Group III | 694 | 7.04 | 76.59 | No |
| ThzLecRLK-L-type-87 | Group IV | 658 | 6.07 | 73.28 | Plasma membrane |
| ThzLecRLK-L-type-111 | Group V-a | 706 | 5.15 | 78.00 | Plasma membrane |
| ThzLecRLK-L-type-109 | Group VI-e | 665 | 4.97 | 74.01 | Plasma membrane |
| ThzLecRLK-C-type-1 | – | 580 | 9.24 | 65.76 | Plasma membrane |
| ThzLecRLK-C-type-2 | – | 541 | 9.06 | 60.45 | Plasma membrane Chloroplast |

each other (*Yang et al., 2016*). Therefore, it can be concluded that the LecRLK sequences of different species differ greatly.

## Conserved domain analysis

Through the domain analysis, we found that there are more G-type ThzLecRLKs than L-type and C-type proteins. It is speculated that G-type ThzLecRLKs may play more diverse roles in plant growth, development, and response to external stimuli. Notably, the majority of ThzLecRLKs (62.96%) contained only one TM domain, 93 contained two TMs, and 17 contained three TM domains. Yang et al.'s findings were similar for poplar (*Yang et al., 2016*). This indicated that LecRLK, as an important membrane binding protein, may bind to the membrane system in a variety of ways. Besides, almost all the genes (16/17) containing three TMs were G-type, which further supports the complexity of the G-type structure.

In order to bind various substrates, the extracellular lectin domains of the LecRLK family are highly variable. However, it can be inferred that, except those responsible for substrate binding specificity, some regions of the ectodomains are conserved among different members of the same family. Consistent with this hypothesis, several highly-conserved motifs were identified within the ectodomains of both G-type and L-type ThzLecRLKs. Two C-rich motifs were identified in the C-terminal region of the EGF motif and the middle of the PAN domain of G-type ThzLecRLKs, similar other species (*Yang et al., 2016*; *Liu et al., 2018*; *Teixeira et al., 2018*). The EGF domain is predicted to be involved in the formation of disulfide bonds and the PAN domain is believed to be involved in protein-protein and protein-carbohydrate interactions (*Naithani et al., 2007*). These conversed motifs of ThzLecRLKs may serve as potential sites of protein-protein interactions. Other conserved motifs may be essential sites for protein activity (*Teixeira et al., 2018*).

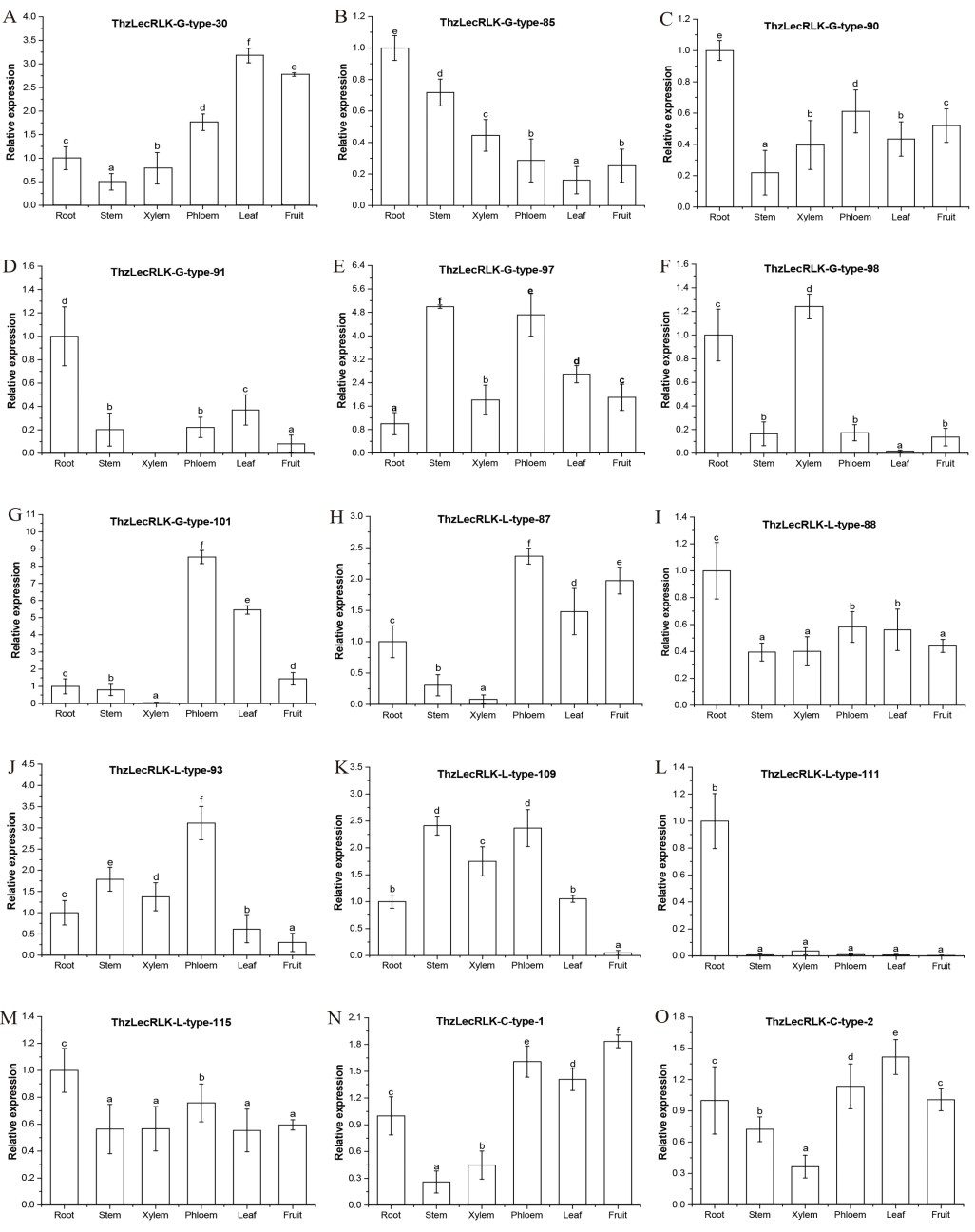

**Figure 10  Expression levels of 15 LecRLK genes in T. hybrid 'Zhongshanshan' tissues.** (A–G) represent seven G-type LecRLK genes, (H–M) 6 L-type LecRLK genes, and (N–O) 2 C-type LecRLK genes. qRT-PCR data were normalized using the APRT gene. The relative expression levels of LecRLKs in tissues were quantified against APRT transcript levels using $2^{-\Delta\Delta CT}$. Three replicates were performed, and error bars indicate the standard deviation of three technical replicates. Values with the same letter in the same gene are not significantly different according to Duncan s multiple range tests ($P < 0.05$).

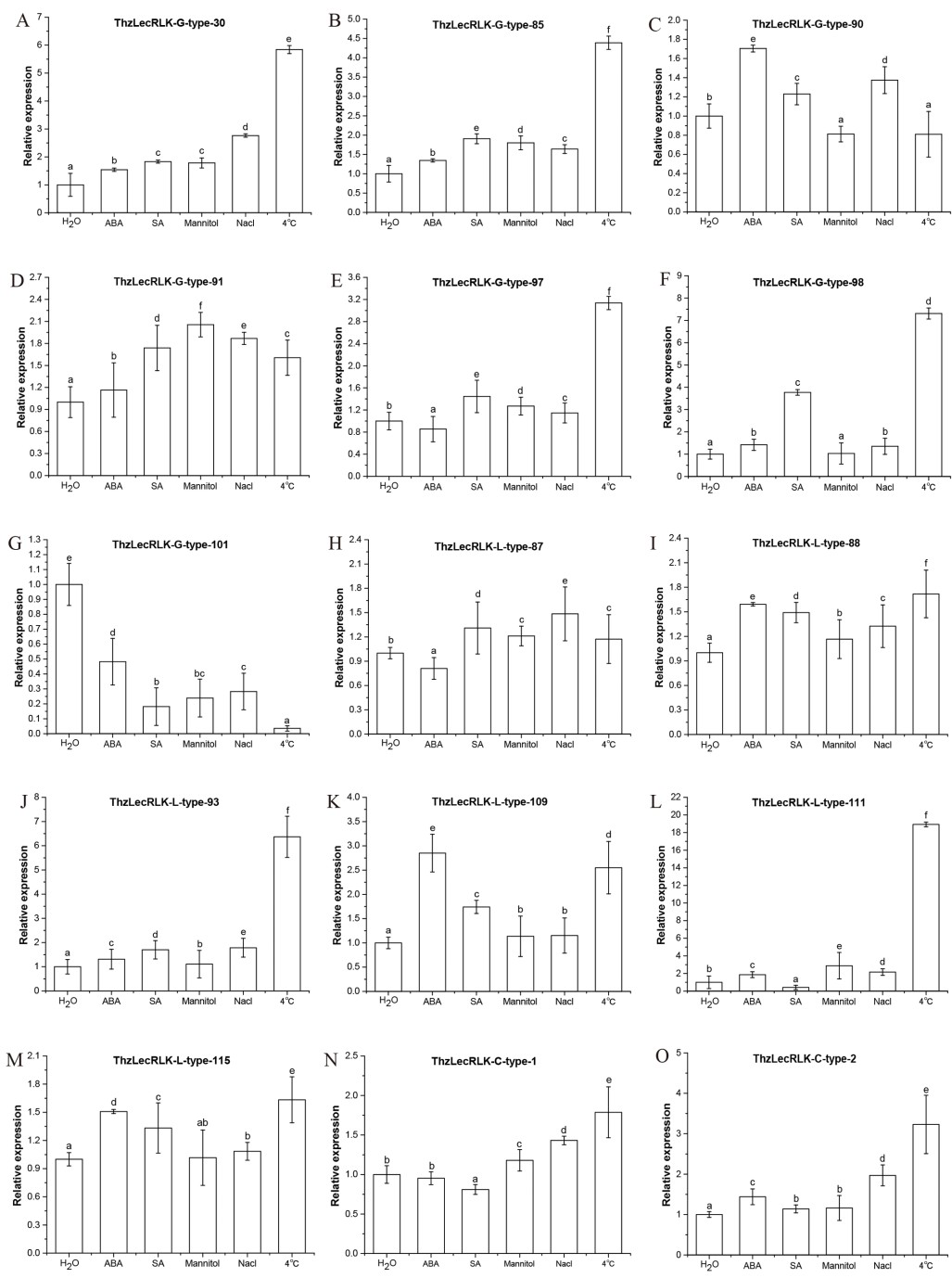

**Figure 11 Expression patterns of 15 selected genes under ABA, SA, mannitol, NaCl and 4 °C treatment.** (A–G) represent seven G-type LecRLK genes, (H–M) 6 L-type LecRLK genes, and (N–O) 2 C-type LecRLK genes. Different letters indicate significant differences at $P < 0.05$ according to Duncan's multiple range tests. qRT-PCR data were normalized using the APRT gene. Three replicates were performed, and error bars indicate the standard deviation of three technical replicates. Values with the same letter in the same gene are not significantly different according to Duncan s multiple range tests ($P < 0.05$).

## Prediction of subcellular localization

Understanding the cellular locations of LecRLK expression is conducive to the analyses of function. Therefore, we predicted the subcellular localization of 297 ThzLecRLKs. The localization results showed that most ThzLecRLKs were located on the plasma membrane, but some were located on chloroplast, vacuole, nucleus, cytoplasm, mitochondrion, extrocytoplasmic surface and endocytoplasmic reticulum. Of the eight loci, all are membranous organelles, except the cytoplasm and extrocytoplasmic surface. Plasma membrane, vacuole, endocytoplasmic reticulum are monolayer organelles, while chloroplast, nucleus and mitochondrion are double membrane organelles (*Bigay & Antonny, 2012*). Based on the results of the prediction of the subcellular localization and transient expression experiments, we speculated that the expression sites of ThzLecRLK family members as membrane binding proteins were not limited to the cell membrane but the entire cell membrane system. This may also be one of the reasons that ThzLecRLKs have more than one TM domain. In addition, ThzlecRLK-C-type-2 and ThzlecRLK-L-type-86 both had multiple locations within the cell, indicating that a considerable number of ThzLecRLKs might be expressed not only in one location, but also in the whole membrane system.

## Gene expression analysis

The patter of gene expression is often suggestive of a gene's biological function. RNA-sequencing and qRT-PCR were combined to reveal the expression patterns of *ThzLecRLKs*. Although 297 *ThzLecRLKs* were identified, the majority (70.71%) of the genes showed limited levels of expression under salt and waterlogging stress. This result was consistent with *Populus*, in which 78 out of 231 *PtLecRLKs* had limited expression in 24 different samples collected under control and treatment conditions (*Yang et al., 2016*). Thus, it can be concluded that although there is a large number of *LecRLKs*, only some may exert biological functions under different space–time conditions.

Some *ThzLecRLK* genes in the same group on phylogenetic tree were found to have a similar expression pattern, suggesting possible functional redundancy of genes within a cluster. For example, three tightly clustered genes (*ThzLecRLK-G-type-29*, *ThzLecRLK-G-type-30*, and *ThzLecRLK-G-type-115*) from G-type Group III-b were also closely clustered on the salt stress heat map. Conversely, some genes had different expression patterns from others in the same cluster, suggesting functional divergence within a subfamily. For example, three genes (*ThzLecRLK-G-type-59*, *ThzLecRLK-G-type-117*, and *ThzLecRLK-G-type-54*) from the G-type Group V-c were significantly induced by salt stress, while the other five genes of the same group were undetectable. qRT-PCR of 15 representative genes from different groups found that there was no obvious regularity in the expression of these representative genes in different tissues and under different stressors, which may be caused by their structural differences, suggesting functional divergence between different clusters.

Both the RNA-sequencing and qRT-PCR analysis showed that partial *LecRLK* genes could be up-regulated or down-regulated by multiple stressors. This is common for *AtLecRLKs*. For example, L-type *LecRK-VI.2/LecRKA4.1*, which played a role in the ABA stress response during seed germination (*Jiang & Yu, 2009*), had upregulated expression

after infection with *Fusarium oxysporum* (*Zhu et al., 2013*), and also contributed to resistance against *Pseudomonas syringae* and *P. carotovorum* (*Singh et al., 2012*). Some *ThzLecRLKs* were found to be stress-specific. Thus, *ThzLecRLK* genes acted on both the specific signal transduction pathways of different stressors and the intersections of different stress signal transduction networks.

qRT-PCR analysis revealed that seven genes were most highly expressed in roots compared with other tissues. In poplar, besides flower bud tissue, the root was found to have the most tissue-specific expression of *LecRLK* genes (*Yang et al., 2016*). Expression may be high in the roots because many of the major stressors plants suffer from are due to changes in soil conditions, such as drought, waterlogging, salt, pathogens, etc., and roots are the first organs coming in contact with these conditions. Twelve of the 15 representative genes had their peak expression levels under low temperature stress. This may be due to the fact that samples (leaves and stems) used for stress analyses were collected from aboveground tissues. Cold stress signal can be directly perceived by plant leaves and stems, while other stress signals are first sensed by roots and then transferred to other tissues. Therefore, the *LecRLK* genes in the sample may sense cold stress earlier.

## CONCLUSIONS

In summary, 297 *LecRLKs* genes were identified for the first time in the transcriptome of *Taxodium* 'Zhongshanshan', including 155 G-type, 140 L-type, and two C-type genes. Evolutionary, structural, and expression analyses suggested divergence of *ThzLecRLK* groups and functional redundancy of the members in the same group. The results of this study shed light on the evolution and function of *ThzLecRLK*, and provide a framework for further functional investigation of these genes.

## ACKNOWLEDGEMENTS

We thank International Science Editing for editing this manuscript.

### Funding

This research was funded by the Jiangsu Agriculture Science and Technology Innovation Fund [No. CX(16)1005], the National Natural Science Foundation of China (31700588), the Natural Science Foundation of Jiangsu (BK20160601) and the National Natural Science Foundation of China (31870592). The funders had no role in study design, data collection and analysis, decision to publish, or preparation of the manuscript.

### Grant Disclosures

The following grant information was disclosed by the authors:
Jiangsu Agriculture Science and Technology Innovation Fund: CX(16)1005.
National Natural Science Foundation of China: 31700588.
Natural Science Foundation of Jiangsu: BK20160601.
National Natural Science Foundation of China: 31870592.

## Competing Interests

Mingzhi Li is employed by Genepioneer Biotechnologies Co. Ltd.

## Author Contributions

- Jinbo Guo performed the experiments, analyzed the data, prepared figures and/or tables, authored or reviewed drafts of the paper, approved the final draft.
- Hao Duan contributed reagents/materials/analysis tools, prepared figures and/or tables.
- Lei Xuan and Ziyang Wang contributed reagents/materials/analysis tools.
- Jianfeng Hua and Chaoguang Yu prepared figures and/or tables.
- Yunlong Yin prepared figures and/or tables, approved the final draft.
- Mingzhi Li analyzed the data.
- Ying Yang conceived and designed the experiments.

## DNA Deposition

The following information was supplied regarding the deposition of DNA sequences:

Sequence information of the 297 ThzLecRLKs is available in GenBank with the accession numbers: MK760259–MK760555.

## Data Availability

The raw data are available in File S1. The data shows the predicted open reading frame and amino acid sequences of all the lecRLK genes identified. These sequences were used for structure and expression analysis.

## Supplemental Information

Supplemental information for this article can be found online at http://dx.doi.org/10.7717/peerj.7498#supplemental-information.

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
