# Peer review of "Identification and functional analysis of LecRLK genes in Taxodium ‘Zhongshanshan’"

_PeerJ, doi:10.7717/peerj.7498_

## Round 0.1 · original submission · Major Revisions

Dear author,

Your paper has been assessed by three reviewers and myself as academic Editor.

As you could see below, the manuscript needs a major revision.
Please address all concerns of the three reviewers and submit a revised version of the manuscript. Please include a detailed response to each reviewer.

English language should be corrected by a native speaker with scientific expertise.

Reviewer 1 ·

Basic reporting

The paper entitled: Identification and functional analysis of LecRLK genes in Taxodium hybrid ‘Zhongshanshan; in this study, authors performed bioinformatic analysis of ThzLecRLK gene family based on transcriptome data and analyzed the expression of 15 representative LecRLKs in different tissues and under various stresses; however, the data is very speculative, the main of this study was predictions of LecRLK gene family; manuscript need to improve language, figures, etc. For example the figure 7; Expression levels of 15 LecRLK genes in T. hybrid 'Zhongshanshan' tissues, this figure does not corresponds to the tissues. Figures 2 and 3 are illegible.



In Discussion:

“The number of ThzLecRLKs identified here exceeded all the other reported species. Since transcriptome only accounts for small proportions of the whole genome, it is clear that many ThzLecRLKs remained uncharacterized”. It is not clear, please rewrite, What is the relationship between the transcriptome and the uncharacterized genes?


Minor

Hroughout their life cycle: change to: Throughout their life cycle

Experimental design

In results:

Authors concluded that: Subcellular localization prediction found that they were mainly concentrated in the cell membrane system; although I believe that due to the characteristics of the domains of these RLKs, they will be found in membranes, the in silico data to identify them in chloroplast membranes, or other organelles is interesting; However, the authors would have to demonstrate some LecRLKs in plant data with fusions with GFP, etc., because the correlation between the subcellular localization in silico and the in plant has been very low.

Data from RNA-seq-based transcriptomics, these are not clear; In which conditions of salt and waterlogging were carried out, in what times were analyzed?;

Also, data of Unigenes co-expressed with LecRLKs in transcriptome are not clear, for example, how many expression conditions were analyzed to determine which genes are co-expressed.

Validity of the findings

The materials and methods are incomplete.

The manuscript can not be revised, since it is incomplete, the authors do not describe the materials and methods, results, footnotes, concentration of ABA, SA, mannitol, and NaCl, used and how long of treatments, without these data nothing can be concluded.

Reviewer 2 ·

Basic reporting

In this paper the authors identified the 297 putative LecRLK genes from a Taxodium 'Zhongshanshan'(?) and classified the genes to 3 types including 155 G-type, 140 L-type and 2 C-type based on the cladistics analysis of gene sequences. This kind of study is relatively rare in conifers and it worth to be published in PeerJ. But several revisions are required before acceptance of the manuscript to PeerJ.

Experimental design

no comment

Validity of the findings

no comment

Additional comments

I point out two major issues. The first issue can be revise easily. But, the second issue requires reanalysis of data and proper graphic works for revision.

First, the plant name, Taxodium hybrid 'Zhongshanshan', is not acceptable names under either the current botanical nomenclature or the cultivar nomenclature. The name 'Zhongshanshan' looks like a cultivar name because authors used in a single quotation mark. I am not sure the origin of this cultivar. Is it a di-specific hybrid or a tri-specific hybrid? Is it a registered cultivar name? The author should clarify or explain more details about their research material with the information of the hybrid origin. The T. hybrid 'Zhongshanshan' (line 126, 143 and many other places) also not usable name, too. The genus name can’t be abbreviated without accompanying specific epithet. Please consult the right expression of cultivar name to knowledgeable botanist in nomenclatural field.

Second, the authors constructed a cladogram of 297 LecRLK gene sequences using NJ tree reconstruction methods. The cladogram (not phylogram, the term of plylogram throughout manuscript should be change to cladogram because no polarity determination in the tree). Fig. 1 simply show branching patterns of genes. This circle cladoram looks nice to see, but there is no information for the branch lengths. Therefore, the classification of group and subgroups based on the branching patterns are totally invalid. In order to use the trees to classifying the gene groups, the NJ tree with branch lengths should be presented in Fig. 1 rather than the current circular cladogram. Then, the unrooted tree should be oriented using the midpoint rooting method or the longest branch rooting method because there is no outgroup or sister gene groups. The subgroup classification of G-type and L-type in Figure 2 should be re-rooted by the sister groups such as C-type genes. In addition, Fig 3 and 4 also need to re-oriented by proper methods. I strongly recommend to authors to change the trees in Figs. 1 and 2 to the NJ distance trees rather than the branching pattern. The NJ distance tree can be presented with supporting values, too.

Reviewer 3 ·

Basic reporting

The authors report on the bioinformatic identification of putative Lectin receptor-like protein kinases (LecRLK) from the transcriptome of the hybrid species Taxodium hybrid ‘Zhongshanshan’, a water-tolerant tree that it is being used on China for aforestation. LecRLKs have been reported for other species but only for a handful of woody plants. These kinases are important on processes of tolerance to biotic and abiotic stresses as signal transductors. In addition to the identification of the putative LecRLKs, the authors also searched on the transcriptome for co-expressed transcription factors (TFs) that may be associated to the LecRLKs and test the expression patterns of 15 LecRLKs on ABA, salicylic acid, mannitol, NaCl, and low temperature.

This manuscript is well organized and details the methodological plan followed.

To the best of my knowledge, the Introduction and background cover the relevant literature.

There are some methodological problems that detract from the study on its present form:

1. Although the HMM profiles are a powerful method of identifying putative proteins, its results need to be curated before further analyzed. HMM profiles identify proteins by a probabilistic metric so sometimes proteins that do not contain full domains can be reported as valid results.
2. The subcellular localization analysis needs to be complemented with at least a second prediction tool.
3. The manuscript should be thoroughly proof-read to improve its English quality. Just to cite a couple of problems, past and present tenses are mixed as well as singular and plurals, and the order of adverbs, adjectives, and nouns.

Experimental design

As noted in the previous section, the methodology is, in general, of sufficient clarity and detail, but care needs to be taken on the reporting of the HMM and PSORT results.
On lines 189 and subsequent, authors indicate that they found (using profile PF00069) 1331 unigenes predicted to contain a protein kinase domain. These unigenes are filtered to remove those without complete ORFs, transmembrane domains, and LecRLK domains. Nevertheless, on lines 226-227 it is stated that at least five L-type ThzLecRLKs lack a kinase domain. Unless this is a missing second kinase domain these putative LecRLKs cannot be such. A diagram of the typical expected G-type, L-type, and C-type LecRLK domains should be included to aid on the understanding of the findings (in addition to figure 3). Also, the resolution of figure 3 is not enough to read the labels.

With respect to the results of the PSORT subcellular localization of the putative LecLRLKs, it has to be noted that this is a prediction tool that is not 100% accurate. I would recommend using at least a second localization tool (for example, http://www.jci-bioinfo.cn/pLoc_bal-mEuk/) and report the results that agree for both tools.

Although most of the methodology is very detailed, there are some omissions. An important one is regarding the construction of the multiple alignment and the phylogenetic tree. The authors stated that they used the MAFFT algorithm (which is a very good choice) but neglected to inform which of its modes was used. Given the nature of the different domain architectures of the L, G and C RLKs, the best choice would be to use the E-INS-I mode (suitable for multiple conserved domains and long gaps). If this mode wasn’t used the multiple alignment could contain mistakenly aligned domains that would impact on the phylogenetic tree. I would suggest the authors include, as supplementary material, the full alignment on a suitable text format (FASTA, clustal, etc.).

My last suggestion is about the presentation of results on the discussion. On lines 329-339 the authors compare the number of LecRLKs found in this study to previous ones. It would be useful if these numbers could be related to the number of proteins and/or genome sizes of the mentioned species. The authors actually mention that Taxodium has a large genome size (20Gb). Maybe presenting these numbers as a percentage of estimated protein-coding genes could aid on visualizing the expansion of these kinases (since C-type LecRLKs still remain very limited).

Validity of the findings

Most of the findings presented are supported by the methodology and results. Care needs to be taken to filter the putative LecRLKs and the corresponding subcellular localizations. If the multiple alignment is redone using the E-INS-I mode and the phylogenetic tree groupings change, these changes should be taken into consideration for the results and discussions.

---

## Round 0.2 · accepted · Accept

Dear author

I can read that you have addressed all the reviewers concerns. The reviewers comments have been responded adequately.

A final check by Gerard Lazo, one of our Section Editors, has resulted in the following request:

"Considering that the authors are trying to characterize a rather large gene family it is requested that the assignment of GO terms be applied to the data. There were some KEGG assignments made; however, the connection of the individual sequence accessions to the terms was not evident. Such assignment should also be made for the GO terms, and done so in one of the table listings. Journal manuscripts are often scanned by text-mining software that locates and extracts core data elements, like gene function. Adding standard ontology terms, such as the Gene Ontology (GO, geneontology.org) or others from the OBO foundry (obofoundry.org) can enhance the recognition of your contribution and description. This will also make human curation of literature easier and more accurate. None of this was visible. As these sequences are tied to specific expression patterns in tissue-types and conditions, the appropriate GO terminology should be applied."

Please address this request in a final revision.